# Partially Dedifferentiated Primitive Malignant Melanoma with Pseudo-Angiomatous Features: A Case Report with Review of the Literature

**DOI:** 10.3390/diagnostics13030495

**Published:** 2023-01-29

**Authors:** Francesca Ambrogio, Anna Colagrande, Eliano Cascardi, Mauro Grandolfo, Raffaele Filotico, Caterina Foti, Carmelo Lupo, Nadia Casatta, Giuseppe Ingravallo, Gerardo Cazzato

**Affiliations:** 1Section of Dermatology and Venereology, Department of Precision and Regenerative Medicine and Ionian Area (DiMePRe-J), University of Bari “Aldo Moro”, 70124 Bari, Italy; 2Section of Molecular Pathology, Department of Emergency and Organ Transplantation (DETO), University of Bari “Aldo Moro”, 70124 Bari, Italy; 3Department of Medical Sciences, University of Turin, 10124 Turin, Italy; 4Pathology Unit, FPO-IRCCS Candiolo Cancer Institute, Str. Provinciale 142 lm 3.95, 10060 Candiolo, Italy; 5Innovation Department, Diapath S.p.A., Via Savoldini n.71, 24057 Martinengo, Italy

**Keywords:** angiomatoid, malignant melanoma, dedifferentiated, undifferentiated, differential diagnosis, vascular neoplasms, malignancies

## Abstract

Malignant melanoma (MM) is traditionally known as the “great mime” of human pathology, as it is potentially capable of imitating the most disparate neoplasms. It is known that in addition to the more classic histotypes of MM, there are also rare forms, including angiomatoid MM. Similarly, it has been amply demonstrated in the literature that MM is capable of dedifferentiating, losing melanocytic lineage markers, constituting a diagnostic challenge for the pathologist. Although 5 cases of primary angiomatoid MM have been described in the literature, to the best of our knowledge, no cases of dedifferentiated melanoma with pseudo-angiomatoid aspects have ever been described. In this paper, we present a very rare case of partially dedifferentiated MM in which the most dedifferentiated component lost melanocytic lineage immunohistochemical markers and assumed a pseudo-angiomatous morphology. Given the rarity of the case, we carried out a literature review of similar cases described, trying to draw new future perspectives not only about this particular variant of MM but also about the widest field of dedifferentiation/undifferentiation of MM.

Malignant melanoma (MM) is traditionally known as the “great mime” of human pathology, as it is potentially capable of imitating the most disparate neoplasms [1]. 

Alongside the more frequent forms of MM encountered in daily dermatopathological practice such as superficial spreading type melanoma (SSM) [2], lentigo maligna melanoma (LMM) [3], nodular melanoma (NM) [4], acral type lentiginous melanoma (ALM) [5,6], and mucosal melanoma [7], there are rarer and more bizarre forms including desmoplastic melanoma (DM) [8], nevoid melanoma (NEM) [9,10] and, among the even rarer forms, the angiomatoid form of MM has been described [11,12]. Furthermore, it has been amply demonstrated in the literature that MM is able to dedifferentiate, partially or totally losing the melanocyte differentiation markers such as Melan-A, HMB-45, tyrosinase, MITF, and others [12,13], posing major diagnostic challenges in the field of differential diagnosis with occult neoplasms [14]. More in detail, the dedifferentiation of melanoma cells constitutes a key point of the morpho-phenotypic plasticity of MM, as well as of its ability to and become resistant to currently used therapies, such as immunotherapy and molecularly targeted therapies [15]. In this field, for example, Chung J. et al. reported a case of a 72-year-old patient with a previously resected melanoma in situ (MIS) who, after 5 years, had MIS relapse with a malignant sarcomatoid component of dedifferentiated melanoma, expressing no immunohistochemical markers other than CD10, a very non-specific marker [16].

Although five cases of primary angiomatoid MM have been described in the literature, to the best of our knowledge, no cases of dedifferentiated melanoma with pseudo-angiomatoid aspects have ever been described. 

In this paper, we present a very rare case of partially dedifferentiated MM in which the most dedifferentiated component lost melanocytic lineage immunohistochemical markers and assumed a pseudo-angiomatous morphology, without expression of vascular immunohistochemical markers. Given the rarity of the case, we carried out a literature review of similar cases described, trying to draw new future perspectives not only about this particular variant of MM but also about the widest field of dedifferentiation/undifferentiation of MM.

The patient we present was an 87-year-old man, in good general condition, who, in his medical history, reported the appearance of a nodular bleeding greyish lesion with irregular borders, on erythematous skin, present for two years (Figure 1). By virtue of the clinical and dermoscopical features, surgical removal with large resection margins was performed, and the sample was sent to the pathological anatomy laboratory, with the clinical suspicion of MM or squamous cell carcinoma (SCC). After fixation in 10% buffered formalin, sampling, processing, and inclusion in paraffin, sections of about 5 microns thickness were prepared, stained with hematoxylin–eosin (H&E) of routine, and further sections were kept for immunostaining with antibodies for melanocytic markers.

The following antibodies (clone, manufacturing company, dilution) were used for immunohistochemical analysis: Anti-S-100 protein (Polyclonal, Dako, Santa Clara, CA, USA, 1:500), Anti-SOX-10 (Polyclonal, ThermoFisher, Bush Road. Albany, NY, USA, 1:2000), Anti-Melan-A (M2-7C10 + M2-9E3, ThermoFisher, 1:100), Anti-HMB-45 (ThermoFisher, 1:100), Anti-CD34 (QBEND/10, Dako, 1:100), Anti-CD31 (JC70A, Dako, 1:40), Anti-ERG (SP06-04, ThermoFisher, 1:100), Anti-Podoplanin (D2-40, ThermoFisher, 1:40).

Furthermore, the authors conducted a literature review following the Preferred Reporting Items for Systematic Reviews and Meta-Analyses (PRISMA) guidelines [17] using PubMed and Web of Sciences (WoS) as main databases and using the keywords: “Angiomatoid Malignant Melanoma” or “Pseudo-angiomatoid Malignant Melanoma” or “Dedifferentiated Melanoma” and “histology” and/or “histopathology”. During the review process case reports, case series, reviews, original articles, editorials, and letters to the editor were included in the study, and English-language articles published up to 23/12/2022 were included.

Histopathological analysis showed pleomorphic and atypical elements, with numerous typical and atypical mitotic figures, eosinophilic intracytoplasmic paranuclear inclusions, nuclei with thinned chromatin, and numerous central and peripheral nucleoli (Figure 2A,B). 

Furthermore, no pigment interspersed with this cell population was appreciated (Figure 2C), but there were large blood-filled spaces bordered by neoplastic cells, not true endothelial cells (c.d. pseudo-vascular pattern) (Figure 2D). 

The following findings were appreciated on immunohistochemical examination: the differentiated component was almost entirely positive for S-100, Melan-A, and HMB-45, but the dedifferentiated component was totally negative for the above markers (Figure 3A); immunohistochemical staining for SOX-10 was almost entirely positive in both components (Figure 3B). Interestingly, the neoplastic proliferation index (Ki67+) was about 5–6% in the differentiated component (Melan-A +, HMB-45 +, S-100 +) and about 20% in the dedifferentiated component (Melan-A -, HMB-45 -, S-100 -) with pseudo-angiomatous aspects.

Further immunohistochemical vascular markers such as CD31, CD34, and ERG were totally negative.

The morphological and immunohistochemical pictures were consistent with the diagnosis of partially dedifferentiated primary malignant melanoma with widespread pseudo-angiomatoid aspects, with Breslow thickness of 7,0 mm, number of mitosis/mmq: 4, presence of vascular invasion, absence of neurotropism and microsatellitosis.

Whole-body positron emission tomography-computed tomography (PET-CT) was performed without any metastatic localization of MM. Furthermore, genetic testing for both V600E and V600K mutations of the BRAF gene was performed using real-time PCR (BRAF RGQ PCR Kit, QIAGEN) revealing a V600E mutation. 

After that, the patient was sent to the O.U. of Plastic and Reconstructive Surgery where he underwent sentinel lymph node biopsy (BLS), which is currently in progress.

Regarding the literature review, a total of 13 articles from both analyzed databases were searched, and, finally, only 5 articles in English were considered suitable for the chosen inclusion criteria. Table 1 summarizes the clinical/pathological findings of the analyzed cases.

Primary angiomatoid melanoma was first described by Adler et al. [18] in 1997 in a patient with cutaneous metastatic melanoma of unknown origin, who presented with a lesion of a metastatic nature on the forehead with a histopathologic pattern reminiscent of a vascular lesion: in fact, pseudo-vascular spaces lined with frankly atypical, pleomorphic cells with erythrocytes were described. Immunohistochemical markers, however, were negative for common vascular markers such as CD31, CD34, and ERG and positive for melanocyte lineage markers. Since that case, five more cases have been described in the literature, including two in the metastatic setting [19] and three as primary lesions [18,20,21]. Various authors have tried to theorize about the pathogenesis of these histopathological peculiarities, and it seems that two theories are the most widely accepted: the theory of “mechanical stress” of blood vessels following biopsy procedures, which could potentially cause this type of feature, or the concept of “vasculogenic mimicry,” already demonstrated by a study in uveal melanoma [22], in which cancer cells would undergo a state of genetic dedifferentiation and expression of mesenchymal markers, by which the cells would be able to avoid neoplastic control mechanisms. From a purely histological point of view, it is important to know and recognize the presence of an angiomatoid MM as it can potentially be confused with other neoplastic lesions, including angiosarcoma or angiomatoid SCC. The peculiarity of our case, besides the feature of the angiomatoid appearance, is that in our patient, these morphological aspects were realized in the dedifferentiated MM part, and this might be in agreement with the concept of “vasculogenic mimicry” in that the component that lost the melanocytic differentiation markers took on an angiomatoid morphological appearance, although it did not express the vascular markers [23]. Regarding the prognosis of patients, all case reports reported in the literature presented a very poor prognosis, also in accordance with the rather high Breslow thickness.

In conclusion, the angiomatoid morphological aspect constitutes one of the different variants of MM, whose knowledge and recognition are of great importance to avoid misdiagnosis with other entities. Furthermore, the knowledge of the negativity of immunohistochemical markers in cases of MM dedifferentiation is a milestone, as an incorrect interpretation could potentially cause a diagnostic error. The use of SOX-10 is the safest way to ensure a correct histological diagnosis.

## Figures and Tables

**Figure 1 diagnostics-13-00495-f001:**
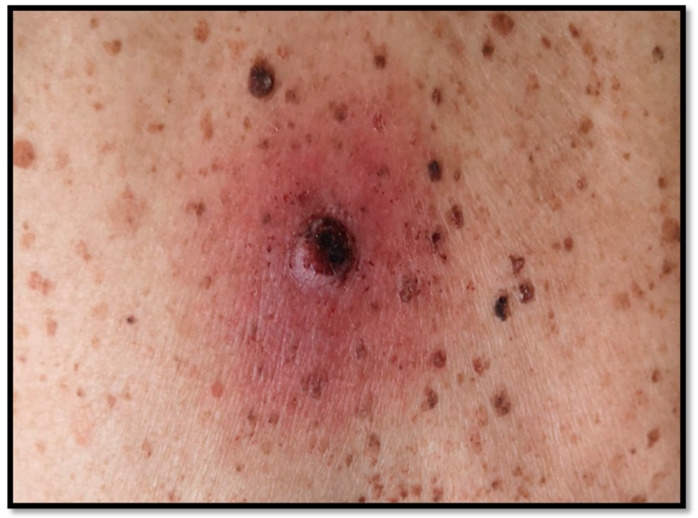
Clinical appearance of the nodular, bleeding lesion with irregular borders.

**Figure 2 diagnostics-13-00495-f002:**
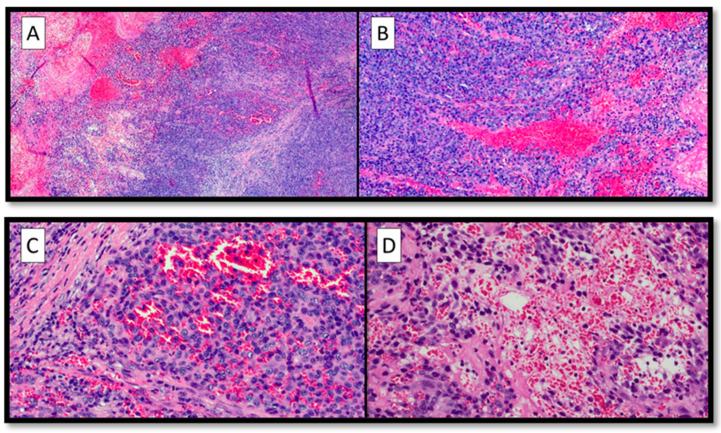
(**A**) Panoramic histopathological view of the lesion examined: note the presence of an abundant pseudo-vascular cell component with extravasated erythrocytes (hematoxylin–eosin, original magnification 4×). (**B**) Histological preparation showing cytological details of the neoplastic cells: pleomorphic, atypical elements, with pseudo-vascular appearance (hematoxylin–eosin, original magnification 10×). (**C**) Photomicrographs showing pseudo-vascular structures containing red blood cells, with some foci of necrosis (H&E, original magnification 20×). (**D**) Details of the previous picture showing foci of hemorrhage (H&E, original magnification 20×).

**Figure 3 diagnostics-13-00495-f003:**
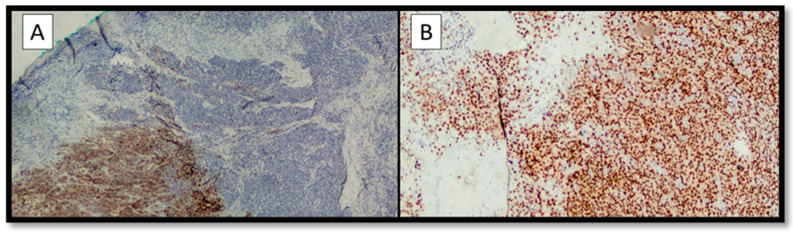
(**A**) Immunohistochemical preparation for Melan-A: note the positivity of Melan-A in the differentiated component (bottom left) and the negativity in the dedifferentiated part of the MM (top right and center) (immunohistochemistry for Melan-A, original magnification 4×). (**B**) Immunohistochemical staining for SOX-10: note the almost total positivity of the marker in all components of the MM (immunohistochemistry for SOX-10, original magnification 10×).

**Table 1 diagnostics-13-00495-t001:** Review of the cases of angiomatoid malignant melanoma previous described in literature.

Number of Case	Sex, Age	Localization	Histological Diagnosis	Immunohistochemical Features
1	M, 44	Intravertebral Metastasis on forehead	MM Angiomatoid metastatic MM	S-100, HMB-45 and vimentin: positivity (+)
2	M, 84	Periorbital region	Primitive DM with angiomatoid pattern	S-100 positivity (+)
3	F, 56	Right arm Metastasis back	MM Angiomatoid MM	S-100 positivity (+), Melan-A and HMB-45 positivity (+)
4	M, 61	Third finger of the left hand Axillary lymph node	MM Metastatic angiomatoid MM	S-100 positivity (+)
5	M, 59	Right thigh	Angiomatoid MM	S-100 positivity (+), HMB-45 positivity (+)
6	F, 63	Right scapular region	Angiomatoid MM	S-100, Melan-A e HMB-45 positivity (+)

*Legend.* DM: desmoplastic melanoma.

## Data Availability

Not applicable.

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
