# Peer review of "Partially Dedifferentiated Primitive Malignant Melanoma with Pseudo-Angiomatous Features: A Case Report with Review of the Literature"

_diagnostics, 2023, doi:10.3390/diagnostics13030495_

Round 1
Reviewer 1 Report
The authors described "Partially dedifferentiated primitive Malignant Melanoma with pseudo-angiomatous features: a case report with review of the literature". As they mentioned, this case should be rare and informative for potential readers. However, I have some concerns about the content as follows.
1. This manuscript is "Case Report". So, they must describe the case in more detail (e.g. How about postoperative course? How about therapy for this case?). Please add it.
2. Please separate case and discussion from main text.
3. How about LYVE1 immnunostaining? Can Malignant Melanoma with pseudo-angiomatous features metastasis through lymphatic vessels?
Author Response
Reviewer n’1: This manuscript is "Case Report". So, they must describe the case in more detail (e.g. How about postoperative course? How about therapy for this case?). Please add it.
Answer n’1: Dear Reviewer n’1, thank you very much for these useful tips to improve the quality of our manuscript. We, currently, add some informations about our patient. Thanks again.
Reviewer n’1: Please separate case and discussion from main text.
Answer n’2: thank you very much, actually we separated it even though as a format of "interesting image" from Diagnostics it should be kept together.
Reviewer n’1: How about LYVE1 immnunostaining? Can Malignant Melanoma with pseudo-angiomatous features metastasis through lymphatic vessels?
Answer n’3: Thank you very much, actually we performed it as an immunohistochemical marker but it turned out to be negative, as this type of MM has no progression characteristics other than common melanoma histotypes, with no particular tropism for lymphatic vessels. Thank you for everything.
Reviewer 2 Report
Delate Figure on Table 4-it is redundant with text and not necessary.
Comment: as a non-Pathologist this does not sounds like a very challenging case-the differentiated components were strongly positive for all melanoma markers.
Author Response
Dear Reviewer n'2, thank you very much. We removed the table of the figure 4.
A warm greeting
Round 2
Reviewer 1 Report
The authors revised the article precisely.